# Privacy-Preserving CNN Training with Transfer Learning

## Abstract

Privacy-preserving nerual network inference has been well studied while homomorphic CNN training still remains an open challenging task. In this paper, we present a practical solution to implement privacy-preserving CNN training based on mere Homomorphic Encryption (HE) technique. To our best knowledge, this is the first attempt successfully to crack this nut and no work ever before has achieved this goal. Several techniques combine to accomplish the task:: (1) with transfer learning, privacy-preserving CNN training can be reduced to homomorphic neural network training, or even multiclass logistic regression (MLR) training; (2) via a faster gradient variant called `Quadratic Gradient`, an enhanced gradient method for MLR with a state-of-the-art performance in convergence speed is applied in this work to achieve high performance; (3) we employ the thought of transformation in mathematics to transform approximating Softmax function in the encryption domain to the approximation of the Sigmoid function. A new type of loss function termed `Squared Likelihood Error` has been developed alongside to align with this change.; and (4) we use a simple but flexible matrix-encoding method named `Volley Revolver` to manage the data flow in the ciphertexts, which is the key factor to complete the whole homomorphic CNN training. The complete, runnable C++ code to implement our work can be found at: `https://anonymous.4open.science/r/HE-CNNtraining-B355/`.

We select `REGNET_X_400MF` as our pre-trained model for transfer learning. We use the first 128 MNIST training images as training data and the whole MNIST testing dataset as the testing data. The client only needs to upload 6 ciphertexts to the cloud and it takes $\sim$ 21 mins to perform 2 iterations on a cloud with 64 vCPUs, resulting in a precision of 21.49%.

## 1 Introduction

### 1.1 Background

Applying machine learning to problems involving sensitive data requires not only accurate predictions but also careful attention to model training. Legal and ethical requirements might limit the use of machine learning solutions based on a cloud service for such tasks. As a particular encryption scheme, homomorphic encryption provides the ultimate security for these machine learning applications and ensures that the data remains confidential since the cloud does not need private keys to decrypt it. However, it is a big challenge to train the machine learning model, such as neural networks or even convolution neural networks, in such encrypted domains. Nonetheless, we will demonstrate that cloud services are capable of applying neural networks over the encrypted data to make encrypted training, and also return them in encrypted form.

Submitted to 37th Conference on Neural Information Processing Systems (NeurIPS 2023). Do not distribute.

## 1.2 Related work

Several studies on machine learning solutions are based on homomorphic encryption in the cloud environment. Since Gilad-Bachrach et al. [1] firstly considered privacy-preserving deep learning prediction models and proposed the private evaluation protocol `CryptoNets` for CNN, many other approaches [2, 3, 4, 5] for privacy-preserving deep learning prediction based on HE or its combination with other techniques have been developed. Also, there are several studies [6, 7, 8, 9] working on logistic regression models based on homomorphic encryption.

However, to our best knowledge, no work ever before based on mere HE techique has presented an solution to successfully perform homomorphic CNN training.

## 1.3 Contributions

Our specific contributions in this paper are as follows:

1. with various techniques, we initiate to propose a practical solution for privacy-preserving CNN training, demonstrating the feasibility of homomorphic CNN training.

2. We suggest a new type of loss function, `Squared Likelihood Error` (SLE), which is friendly to pervacy-perserving manner. As a result, we can use the Sigmoid function to replace the Softmax function which is too diffuclt to calculate in the encryption domain due to its uncertainty.

3. We develop a new algorithm with SLE loss function for MLR using `quadratic gradient`. Experiments show that this HE-friendly algorithm has a state-of-the-art performance in convergence speed.

## 2 Preliminaries

We adopt "$\otimes$" to denote the kronecker product and "$\odot$" to denote the component-wise multiplication between matrices.

### 2.1 Fully Homomorphic Encryption

Homomorphic Encryption (HE) is one type of encryption scheme with a special characteristic called $Homomorphic$, which allows to compute on encrypted data without having access to the secret key. Fully HE means that the scheme is fully homomorphic, namely, homomorphic with regards to both addition and multiplication, and that it allows arbitrary computation on encrypted data. Since Gentry proposed the first fully HE scheme [10] in 2009, some technological progress on HE has been made. For example, Brakerski, Gentry and Vaikuntanathan [11] present a novel way of constructing leveled fully homomorphic encryption schemes (BGV) and Smart and Vercauteren [12] introduced one of the most important features of HE systems, a packing technique based on polynomial-CRT called Single Instruction Multiple Data (aka SIMD) to encrypt multiple values into a single ciphertext. Another great progress in terms of machine learning applications is the $rescaling$ procedure [13], which can manage the magnitude of plaintext effectively.

Modern fully HE schemes, such as `HEAAN`, usually support seveal common homomorphic operations: the encryption algorithm `Enc` encrypting a vector, the decryption algorithm `Dec` decrypting a ciphertext, the homomorphic addition `Add` and multiplication `Mult` between two ciphertexts, the multiplication `cMult` of a contant vector with a ciphertext, the rescaling operation `ReScale` to reduce the magnitude of a plaintext to an appropriate level, the rotation operation `Rot` generating a new ciphertext encrypting the shifted plaintext vector, and the bootstrapping operation `bootstrap` to refresh a ciphertext usually with a small ciphertext modulus.

### 2.2 Database Encoding Method

For a given database $Z$, Kim et al. [6] first developed an efficient database encoding method, in order to make full use of the HE computation and storage resources. They first expand the matrix database to a vector form $V$ in a row-by-row manner and then encrypt this vector $V$ to obtain a ciphertext $Z = Enc(V)$. Also, based on this database encoding, they mentioned two simple operations via

shifting the encrypted vector by two different positions, respectively: the complete row shifting and the *incomplete* column shifting. These two operations performing on the matrix $Z$ output the matrices $Z^{'}$ and $Z^{''}$, as follows:

$$Z = \begin{bmatrix} x_{10} & x_{11} & \dots & x_{1d} \\ x_{20} & x_{21} & \dots & x_{2d} \\ \vdots & \vdots & \ddots & \vdots \\ x_{n0} & x_{n1} & \dots & x_{nd} \end{bmatrix}, \qquad Z^{'} = Enc \begin{bmatrix} x_{20} & x_{21} & \dots & x_{2d} \\ \vdots & \vdots & \ddots & \vdots \\ x_{n0} & x_{n1} & \dots & x_{nd} \\ x_{10} & x_{11} & \dots & x_{1d} \end{bmatrix},$$

$$Z^{''} = Enc \begin{bmatrix} x_{11} & \dots & x_{1d} & x_{20} \\ x_{21} & \dots & x_{2d} & x_{30} \\ \vdots & \vdots & \ddots & \vdots \\ x_{n1} & \dots & x_{nd} & x_{10} \end{bmatrix}, \quad Z^{'''} = Enc \begin{bmatrix} x_{11} & \dots & x_{1d} & x_{10} \\ x_{21} & \dots & x_{2d} & x_{20} \\ \vdots & \vdots & \ddots & \vdots \\ x_{n1} & \dots & x_{nd} & x_{n0} \end{bmatrix}.$$

The complete column shifting to obtain the matrix $Z^{'''}$ can also be achieved by two `Rot`, two `cMult`, and an `Add`.

Other works [14, 4] using the same encoding method also developed some other procedures, such as `SumRowVec` and `SumColVec` to calculate the summation of each row and column, respectively. Such basic common and simple operations consisting of a series of HE operations are significantly important for more complex calculations such as the homomorphic evaluation of gradient.

## 2.3 Convolutional Neural Network

Inspired by biological processes, Convolutional Neural Networks (CNN) are a type of artificial neural network most commonly used to analyze visual images. CNNs play a significant role in image recognition due to their powerful performance. It is also worth mentioning that the CNN model is one of a few deep learning models built with reference to the visual organization of the human brain.

### 2.3.1 Transfer Learning

Transfer learning in machine learning is a class of methods in which a pretrained model can be used as an optimization for a new model on a related task, allowing rapid progress in modeling the new task. In real-world applications, very few researchers train entire convolutional neural networks from scratch for image processing-related tasks. Instead, it is common to use a well-trained CNN as a fixed feature extractor for the task of interest. In our case, we freeze all the weights of the selected pre-trained CNN except that of the final fully-connected layer. We then replace the last fully-connected layer with a new layer with random weights (such as zeros) and only train this layer.

**REGNET_X_400MF**   To use transfer learning in our privacy-preserving CNN training, we adopt a new network design paradigm called `RegNet`, recently introduced by Facebook AI researchers, as our pre-trained model. RegNet is a low-dimensional design space consisting of simple, regular networks. In particular, we apply `REGNET_X_400MF` as a fixed feature extractor and replaced the final fully connected layer with a new one of zero weights. CNN training in this case can be simplified to multiclass logistic regression training. Since `REGNET_X_400MF` only receive color images of size $224 \times 224$, the grayscale images will be stacked threefold and images of different sizes will be resized to the same size in advance. These two transformations can be done by using PyTorch.

### 2.3.2 Datasets

We adopt three common datasets in our experiments: MNIST, USPS, and CIFAR10. Table 1 describes the three datasets.

## 3 Technical details

## 3.1 Multiclass Logistic Regression

Multiclass Logistic Regression, or Multinomial Logistic Regression, can be seen as an extension of logistic regression for multi-class classification problems. Supposing that the matrix $X \in \mathbb{R}^{n \times (1+d)}$,

Table 1: Characteristics of the several datasets used in our experiments

| Dataset | No. Samples (training) | No. Samples (testing) | No. Features | No. Classes |
|---|---|---|---|---|
| USPS | 7,291 | 2,007 | 16×16 | 10 |
| MNIST | 60,000 | 10,000 | 28×28 | 10 |
| CIFAR-10 | 50,000 | 10,000 | 3×32×32 | 10 |

the column vector $Y \in \mathbb{N}^{n \times 1}$, the matrix $\bar{Y} \in \mathbb{R}^{n \times c}$, and the matrix $W \in \mathbb{R}^{c \times (1+d)}$ represent the dataset, class labels, the one-hot encoding of the class labels, and the MLR model parameter, respectively:

$$
X = \begin{bmatrix} x_1 \\ x_2 \\ \vdots \\ x_n \end{bmatrix} = \begin{bmatrix} x_{[1][0]} & x_{[1][1]} & \cdots & x_{[1][d]} \\ x_{[2][0]} & x_{[2][1]} & \cdots & x_{[2][d]} \\ \vdots & \vdots & \ddots & \vdots \\ x_{[n][0]} & x_{[n][1]} & \cdots & x_{[n][d]} \end{bmatrix},
$$

$$
Y = \begin{bmatrix} y_1 \\ y_2 \\ \vdots \\ y_n \end{bmatrix} \xrightarrow{\text{one-hot encoding}} \bar{Y} = \begin{bmatrix} \bar{y}_1 \\ \bar{y}_2 \\ \vdots \\ \bar{y}_n \end{bmatrix} = \begin{bmatrix} y_{[1][1]} & y_{[1][2]} & \cdots & y_{[1][c-1]} \\ y_{[2][1]} & y_{[2][2]} & \cdots & y_{[2][c-1]} \\ \vdots & \vdots & \ddots & \vdots \\ y_{[n][1]} & y_{[n][2]} & \cdots & y_{[n][c-1]} \end{bmatrix},
$$

$$
W = \begin{bmatrix} w_{[0]} \\ w_{[1]} \\ \vdots \\ w_{[c-1]} \end{bmatrix} = \begin{bmatrix} w_{[0][0]} & w_{[0][1]} & \cdots & w_{[0][d]} \\ w_{[1][0]} & w_{[1][1]} & \cdots & w_{[1][d]} \\ \vdots & \vdots & \ddots & \vdots \\ w_{[c-1][0]} & w_{[c-1][1]} & \cdots & w_{[c-1][d]} \end{bmatrix}.
$$

MLR aims to maxsize $L$ or $\ln L$:

$$
L = \prod_{i=1}^{n} \frac{\exp(x_i \cdot w_{[y_i]}^{\mathsf{T}})}{\sum_{k=0}^{c-1} \exp(x_i \cdot w_{[k]}^{\mathsf{T}})} \longmapsto \ln L = \sum_{i=1}^{n} [x_i \cdot w_{[y_i]}^{\mathsf{T}} - \ln \sum_{k=0}^{c-1} \exp(x_i \cdot w_{[k]}^{\mathsf{T}})].
$$

The loss function $\ln L$ is a multivariate function of $[(1+c)(1+d)]$ variables, which has its column-vector gradient $\nabla$ of size $[(1+c)(1+d)]$ and Hessian square matrix $\nabla^2$ of order $[(1+c)(1+d)]$ as follows:

$$
\nabla = \frac{\partial \ln L}{\partial \pi} = \left[ \frac{\partial \ln L}{\partial w_{[0]}}, \frac{\partial \ln L}{\partial w_{[1]}}, \ldots, \frac{\partial \ln L}{\partial w_{[c-1]}} \right]^{\mathsf{T}},
$$

$$
\nabla^2 = \begin{bmatrix} \frac{\partial^2 \ln L}{\partial w_{[0]} \partial w_{[0]}} & \frac{\partial^2 \ln L}{\partial w_{[0]} \partial w_{[1]}} & \cdots & \frac{\partial^2 \ln L}{\partial w_{[0]} \partial w_{[c-1]}} \\ \frac{\partial^2 \ln L}{\partial w_{[1]} \partial w_{[0]}} & \frac{\partial^2 \ln L}{\partial w_{[1]} \partial w_{[1]}} & \cdots & \frac{\partial^2 \ln L}{\partial w_{[1]} \partial w_{[c-1]}} \\ \vdots & \vdots & \ddots & \vdots \\ \frac{\partial^2 \ln L}{\partial w_{[c-1]} \partial w_{[0]}} & \frac{\partial^2 \ln L}{\partial w_{[c-1]} \partial w_{[1]}} & \cdots & \frac{\partial^2 \ln L}{\partial w_{[c-1]} \partial w_{[c-1]}} \end{bmatrix}.
$$

**Nesterov's Accelerated Gradient**    With $\nabla$ or $\nabla^2$, first-order gradient algorithms or second-order Newton–Raphson method are commonly applied in MLE to maxmise $\ln L$. In particular, Nesterov's Accelerated Gradient (NAG) is a practical solution for homomorphic MLR without frequent inversion operations. It seems plausible that the NAG method is probably the best choice for privacy-preserving model training.

### 3.2   Chiang's Quadratic Gradient

Chiang's Quadratic Gradient (CQG) [15, 16, 9] is a faster, promising gradient variant that can combine the first-order gradient descent/ascent algorithms and the second-order Newton–Raphson method, accelerating the raw Newton–Raphson method with various gradient algorithms and probably

135 helpful to build super-quadratic algorithms. For a function $F(x)$ with its gradient $g$ and Hessian
136 matrix $H$, to build CQG, we first construct a diagonal matrix $\bar{B}$ from the Hessian $H$ itself:

$$\bar{B} = \begin{bmatrix} \frac{1}{\varepsilon+\sum_{i=0}^d |\bar{h}_{0i}|} & 0 & \cdots & 0 \\ 0 & \frac{1}{\varepsilon+\sum_{i=0}^d |\bar{h}_{1i}|} & \cdots & 0 \\ \vdots & \vdots & \ddots & \vdots \\ 0 & 0 & \cdots & \frac{1}{\varepsilon+\sum_{i=0}^d |\bar{h}_{di}|} \end{bmatrix},$$

137 where $\bar{h}_{ji}$ is the elements of the matrix $H$ and $\varepsilon$ is a small constant positive number.

138 CQG for the function $F(\mathbf{x})$, defined as $G = \bar{B} \cdot g$, has the same dimension as the raw gradient $g$. To
139 apply CQG in practice, we can use it in the same way as the first-order gradient algorithms, except
140 that we need to replace the naive gradient with the quadratic gradient and adopt a new learning rate
141 (usually by increasing 1 to the original learning rate).

142 For efficiency in applying CQG, a good bound matrix should be attempted to obtain in order to
143 replace the Hessian itself. Chiang has proposed the enhanced NAG method via CQG for MLR with a
144 fixed Hessian [17, 7, 18] substitute built from $\frac{1}{2}X^\intercal X$.

### 3.3 Approximating Softmax Function

146 It might be impractical to perfectly approximate Softmax function in the privacy-preserving domain
147 due to its uncertainty. To address this issue, we employ the thought of transformation from mathemat-
148 ics: transforming one tough problem into another easier one. That is, instead of trying to approximate
149 the Softmax function, we attempt to approximate the Sigmoid function in the encryption domain,
150 which has been well-studied by several works using the least-square method.

In line with standard practice of the log-likelihood loss function involving the Softmax function, we
should try to maximize the new loss function

$$L_1 = \prod_{i=1}^n \frac{1}{1 + \exp(-\mathbf{x}_i \cdot w_{[y_i]}^\intercal)}.$$

151 We can prove that $\ln L_1$ is concave and deduce that $\frac{1}{4}E \otimes X^\intercal X$ can be used to build the CQG for
152 $\ln L_1$. However, the performance of this loss function $\ln L_1$ is not ideal, probably because for the
153 individual example its gradient and Hessian contain no information about any other class weights not
154 related to this example.

`Squared Likelihood Error`   After many attempts to finding a proper loss function, we develop
a novel loss function that can have a competitive performance to the log-likelihood loss function,
which we term `Squared Likelihood Error` (SLE):

$$L_2 = \prod_{i=1}^n \prod_{j=0}^{c-1} (\bar{y}_i - Sigmoid(\mathbf{x}_i \cdot w_{[y_i]}^\intercal))^2 \longmapsto \ln L_2 = \sum_{i=1}^n \sum_{j=0}^{c-1} \ln |\bar{y}_i - Sigmoid(\mathbf{x}_i \cdot w_{[y_i]}^\intercal)|.$$

155 We can also prove that $\ln L_2$ is concave and that $\frac{1}{4}E \otimes X^\intercal X$ can be used to build the CQG for $\ln L_2$.
156 The loss function SLE might be related to Mean Squared Error (MSE): the MSE loss function sums
157 all the squared errors while SLE calculates the cumulative product of all the squared likelihood errors.

158 Combining together all the techniques above, we now have the enhanced NAG method with the SLE
159 loss function for MLR training, described in detail in Algorithm 1.

160 `Performance Evaluation`   We test the convergence speed of the raw NAG method with log-
161 likelihood loss function (denoted as RawNAG), the NAG method with SLE loss function (denoted
162 as SigmoidNAG), and the enhanced NAG method via CQG with SLE loss function (denoted as
163 SigmoidNAGQG) on the three datasets described above: USPS, MNIST, and CIFAR10. Since two
164 different types of loss functions are used in these three methods, the loss function directly measuring
165 the performance of various methods will not be selected as the indicator. Instead, we select precision
166 as the only indicator in the following Python experiments. Note that we use `REGNET_X_400MF` to in

**Algorithm 1** The Enhanced NAG method with the SLE loss function for MLR Training

---

**Input:** training dataset $X \in \mathbb{R}^{n \times (1+d)}$; one-hot encoding training label $Y \in \mathbb{R}^{n \times c}$; and the number $\kappa$ of iterations;

**Output:** the parameter matrix $V \in \mathbb{R}^{c \times (1+d)}$ of the MLR

1:  Set $\bar{H} \leftarrow -\frac{1}{4}X^{\intercal}X$                                  $\triangleright \bar{H} \in \mathbb{R}^{(1+d) \times (1+d)}$

2:  Set $V \leftarrow \mathbf{0}, W \leftarrow \mathbf{0}, \bar{B} \leftarrow \mathbf{0}$         $\triangleright V \in \mathbb{R}^{c \times (1+d)}, W \in \mathbb{R}^{c \times (1+d)}, \bar{B} \in \mathbb{R}^{c \times (1+d)}$

3:  **for** $j := 0$ to $d$ **do**

4:      $\bar{B}[0][j] \leftarrow \varepsilon$                    $\triangleright \varepsilon$ is a small positive constant such as $1e-10$

5:      **for** $i := 0$ to $d$ **do**

6:          $\bar{B}[0][j] \leftarrow \bar{B}[0][j] + |\bar{H}[i][j]|$

7:      **end for**

8:      **for** $i := 1$ to $c - 1$ **do**

9:          $\bar{B}[i][j] \leftarrow \bar{B}[0][j]$

10:      **end for**

11:      **for** $i := 0$ to $c - 1$ **do**

12:          $\bar{B}[i][j] \leftarrow 1.0/\bar{B}[i][j]$

13:      **end for**

14: **end for**

15: Set $\alpha_0 \leftarrow 0.01, \alpha_1 \leftarrow 0.5 \times (1 + \sqrt{1 + 4 \times \alpha_0^2})$

16: **for** $count := 1$ to $\kappa$ **do**

17:      Set $Z \leftarrow X \times V^{\intercal}$               $\triangleright Z \in \mathbb{R}^{n \times c}$ and $V^{\intercal}$ means the transpose of matrix V

18:      **for** $i := 1$ to $n$ **do**          $\triangleright Z$ is going to store the inputs to the Sigmoid function

19:          **for** $j := 0$ to $d$ **do**

20:               $Z[i][j] \leftarrow 1/(1 + e^{-Z[i][j]})$

21:          **end for**

22:      **end for**

23:      Set $\boldsymbol{g} \leftarrow (Y - Z)^{\intercal} \times X$                          $\triangleright \boldsymbol{g} \in \mathbb{R}^{c \times (1+d)}$

24:      Set $G \leftarrow \mathbf{0}$

25:      **for** $i := 0$ to $c - 1$ **do**

26:          **for** $j := 0$ to $d$ **do**

27:               $G[i][j] \leftarrow \bar{B}[i][j] \times \boldsymbol{g}[i][j]$

28:          **end for**

29:      **end for**

30:      Set $\eta \leftarrow (1 - \alpha_0)/\alpha_1, \gamma \leftarrow 1/(n \times count)$          $\triangleright n$ is the size of training data

31:      $w_{temp} \leftarrow W + (1 + \gamma) \times G$

32:      $W \leftarrow (1 - \eta) \times w_{temp} + \eta \times V$

33:      $V \leftarrow w_{temp}$

34:      $\alpha_0 \leftarrow \alpha_1, \alpha_1 \leftarrow 0.5 \times (1 + \sqrt{1 + 4 \times \alpha_0^2})$

35: **end for**

36: **return** $W$

---

advance extract the features of USPS, MNIST, and CIFAR10, resulting in a new same-size dataset with 401 features of each example. Figure 1 shows that our enhanced methods all converge faster than other algorithms on the three datasets.

### 3.4  Double Volley Revolver

Unlike those efficient, complex encoding methods [3], `Volley Revolver` is a simple, flexible matrix-encoding method specialized for privacy-preserving machine-learning applications, whose basic idea in a simple version is to encrypt the transpose of the second matrix for two matrices to perform multiplication. Figure 2 describes a simple case for the algorithm adopted in this encoding method.

The encoding method actually plays a significant role in implementing privacy-preserving CNN training. Just as Chiang mentioned in [4], we show that Volley Revolver can indeed be used to implement homomorphic CNN training. This simple encoding method can help to control and manage the data flow through ciphertexts.

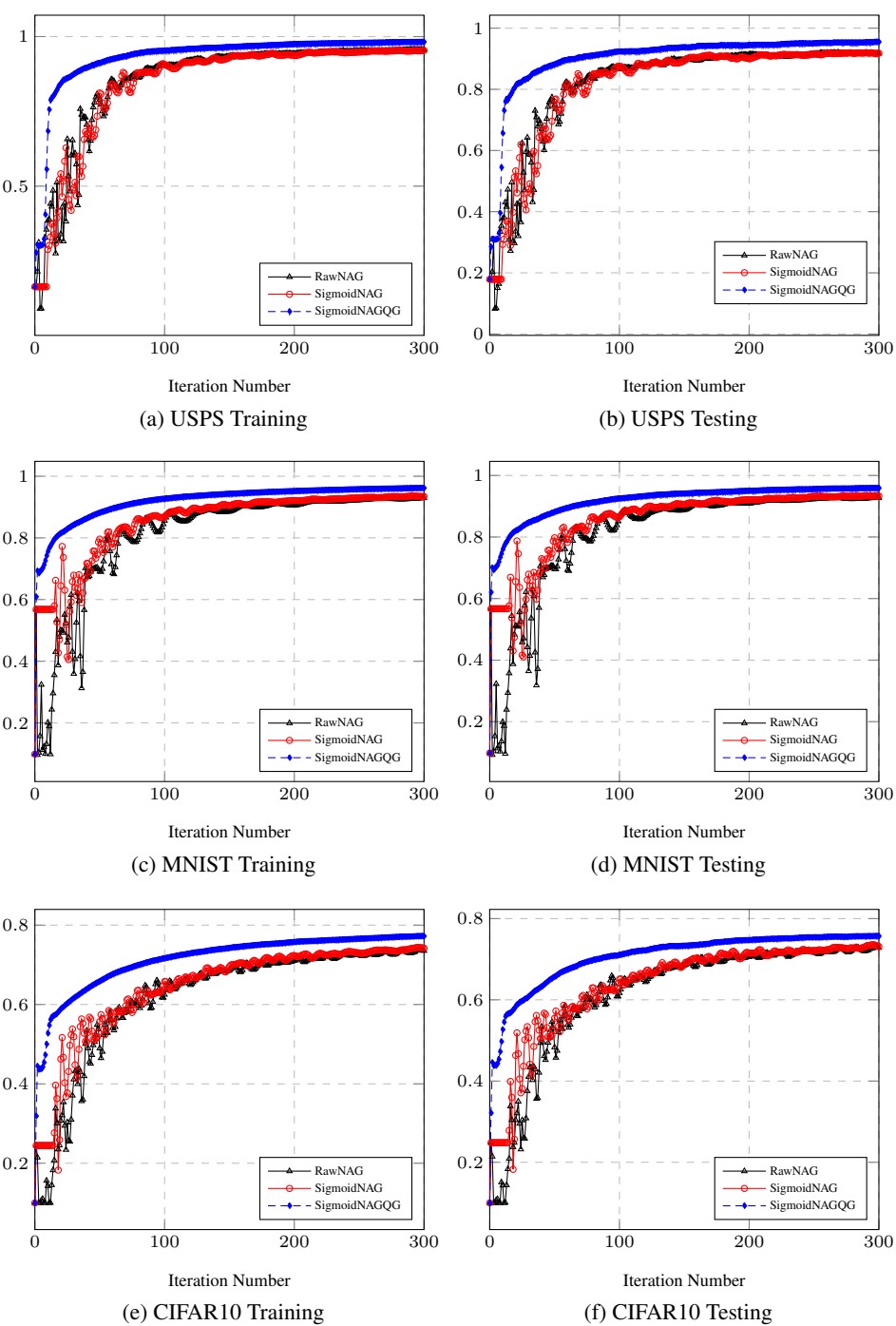

Figure 1: Training and Testing precision results for raw NAG vs. NAG with SLE vs. The enhanced NAG with SLE

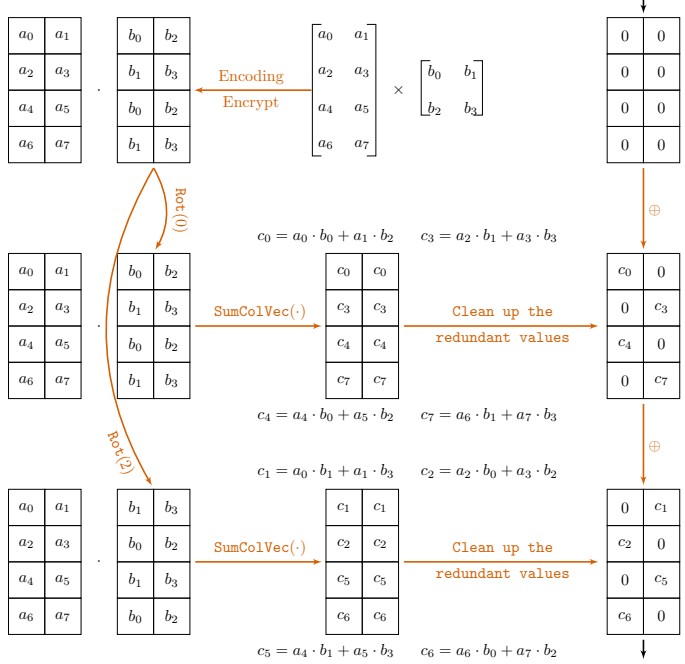

Figure 2: The matrix multiplication algorithm of `Volley Revolver` for the $4 \times 2$ matrix $A$ and the matrix $B$ of size $2 \times 2$

However, we don't need to stick to encrypting the transpose of the second matrix. Instead, either of the two matrices is transposed would do the trick: we could also encrypt the transpose of the first matrix, and the corresponding multiplication algorithm due to this change is similar to the Algorithm 2 from [4].

Also, if each of the two matrices are too large to be encrypted into a single ciphertext, we could also encrypt the two matrices into two teams $A$ and $B$ of multiple ciphertexts. In this case, we can see this encoding method as `Double Volley Revolver`, which has two loops: the outside loop deals with the calculations between ciphertexts from two teams while the inside loop literally calculates two sub-matrices encrypted by two ciphertexts $A_{[i]}$ and $B_{[j]}$ using the raw algorithm of Volley Revolver.

# 4 Privacy-preserving CNN Training

## 4.1 Polynomial Approximation

Although Algorithm 1 enables us to avoid computing the Softmax function in the encryption domain, we still need to calculate the Sigmoid function using HE technique. This problem has been well studied by several works and we adopt a simple one [19], that is (1) we first use the least-square method to perfectly approximate the sigmoid function over the range $[-8, +8]$, obtaining a polynomial $Z_{11}$ of degree 11; and (2) we use a polynomial $Z_3$ of degree 3 to approximate the Sigmoid by minimizing the cost function $F$ including the squared gradient difference:

$$F = \lambda_0 \cdot \int_{-8}^{+8} (Z_{11} - Z_3)^2 dx + \lambda_1 \cdot \int_{-8}^{+8} (Z_{11}' - Z_3')^2 dx,$$

where $\lambda_0$ and $\lambda_1$ are two positive float numbers to control the shape of the polynomial to approximate.

Setting $\lambda_0 = 128$ and $\lambda_1 = 1$ would result in the polynomial we used in our privacy-preserving CNN training: $Z_3 = 0.5 + 0.106795345032 \cdot x - 0.000385032598 \cdot x^3$.

## 4.2 Homomorphic Evaluation

Before the homomorphic CNN training starts, the client needs to encrypt the dataset $X$, the data labels $\bar{Y}$, the matrix $\bar{B}$ and the weight $W$ into ciphertexts $Enc(X)$, $Enc(Y)$, $Enc(\bar{B})$ and $Enc(W)$, respectively, and upload them to the cloud. For simplicity in presentation, we can just regard the whole pipeline of homomorphic evaluation of Algorithm 1 as updating the weight ciphertext: $W = W + \bar{B} \odot (\bar{Y} - Z_3(X \times W^\intercal))^\intercal \times X$, regardless of the subtle control of the enhanced NAG method with the SLE loss function.

Since `Volley Revolver` only needs one of the two matrices to be transposed ahead before encryption and $(\bar{Y} - Z_3(X \times W^\intercal))^\intercal \times X$ happened to suffice this situation between any matrix multiplication, we can complete the homomorphic evaluation of CQG for MLR.

# 5 Experiments

The C++ source code to implement the experiments in this section is openly available at: `https://anonymous.4open.science/r/HE-CNNtraining-B355/`.

**Implementation**  We implement the enhanced NAG with the SLE loss function based on HE with the library `HEAAN`. All the experiments on the ciphertexts were conducted on a public cloud with $64$ vCPUs and 192 GB RAM.

We adopt the first $128$ MNIST training images as the training data and the whole test dataset as the testing data. Both the training images and testing images have been processed in advance with the pre-trained model `REGNET_X_400MF`, resulting in a new dataset with each example of size 401.

## 5.1 Parameters

The parameters of `HEAAN` we selected are: $logN = 16$, $logQ = 990$, $logp = 45$, $slots = 32768$, which ensure the security level $\lambda = 128$. Refer [6] for the details of these parameters. We didn't use bootstrapping to refresh the weight ciphertexts and thus it can only perform 2 iterations of our algorithm. Each iteration takes $\sim 11$mins. The maximum runtime memory in this case is $\sim 18$ GB. The 128 MNIST training images are encrypted into 2 ciphertexts. The client who own the private data has to upload these two ciphertexts, two ciphertexts encrypting the one-hot labels $\bar{Y}$, one ciphertext encrypting the $\bar{B}$ and one ciphertext encrypting the weight $W$ to the cloud. The inticial weight matrix $W_0$ we adopted is the zero matrix. The resulting MLR model after 2-iteration training has reached a pricision of $21.49\%$ and obtain the loss of $-147206$, which are consistent with the Python simulation experiment.

# 6 Conclusion

In this work, we initiated to implement privacy-persevering CNN training based on mere HE techniques by presenting a faster HE-friendly algorithm.

The HE operation bootstrapping could be adopted to refresh the weight ciphertexts. Python experiments imitating the privacy-preserving CNN training using $Z_3$ as Sigmoid substitution showed that using a large amount of data such as 8,192 images to train the MLE model for hundreds of iterations would finally reach $95\%$ precision. The real experiments over ciphertexts conducted on a high-performance cloud with many vCPUs would take weeks to complete this test, if not months.

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
