# OpenReview forum: "Privacy-Preserving CNN Training with Transfer Learning"
_NeurIPS.cc/2023/Conference — Submitted to NeurIPS 2023_

### Official Review · Reviewer_og3N · 2023-06-29

**Soundness:** 4 excellent
**Presentation:** 1 poor
**Contribution:** 3 good
**Rating:** 5
**Confidence:** 3

**Summary:**

This paper combines several existing techniques to achieve privacy-preserving CNN training. These techniques include transfer learning,  Quadratic Gradient,  mathematical transformation, and matrix-encoding method Volley Revolver.

This writing is more of a technical document rather than a research paper with insights.

**Strengths:**

1)For the first time, they apply homomorphic encryption to neural network training.
2)They demonstrate the feasibility of homomorphic CNN training.
3)They propose pervacy-perserving friendly Squared Likelihood Error (SLE) for CNN training.
4)Experimentally, their algorithm has a state-of-the-art performance in convergence speed.

**Weaknesses:**

1)The introduction of related works is pretty simple, which makes it difficult to evaluate the contributes of the paper.
2)The quality of writing/presentation is very weak and unreadable.

**Questions:**

1)Is the propose method suitable for other neural network, like RNN, Transformers?
2)Do you re-evaluate your method by replacing the pretrained REGNET_X_400MF to another one?

**Limitations:**

See Questions and weakness.

---

> ### Author Rebuttal · Authors · 2023-08-09
>
> $\textbf{Response}$
>
> We would like to thank the reviewers for their input. Their comments have been thoroughly considered, and altering the manuscript in accordance with these comments will significantly improve the quality of our paper in the next submission.
>
> **C1: Is the propose method suitable for other neural network, like RNN, Transformers?
>
> A1: The proposed method aims for the MLR model, rather than for other neural networks. We wonder if RNN and Transformers can be reduced to MLR, with transfer learning.
>
> We would like to point out here that the new loss function (so-called SLE) fails to be used in neural networks with hidden layers, in which cases their loss functions might not be concave.
>
> **C2: Do you re-evaluate your method by replacing the pretrained REGNET_X_400MF to another one?
>
> A2: We think we evaluated our proposed method with the pre-trained REGNET_Y_400MF model and perhaps other pre-trained REGNET models. We just chose a pre-trained model from the REGNET family that has the last FC layer with a smaller number of nodes. We definitely would evaluate the proposed method on more pre-trained models other than the REGNET series.
>
> **C3: The introduction of related works is pretty simple, which makes it difficult to evaluate the contributes of the paper.
>
> A3: Yes, we did little survey about the related work part. This writing does seem like a technical document. It is too bold for us to claim this work to be the first without a full survey of the current techniques.

---

> > ### Comment · Reviewer_og3N · 2023-08-12
> >
> > Thank you very much for your detailed explanation of my concerns. I keep my initial score.

---

> > > ### Author Response · Authors · 2023-08-14
> > > **Response to reviewer og3N**
> > >
> > > You're welcome and thank you for your thoughtful consideration.

---

### Official Review · Reviewer_ZXqT · 2023-06-30

**Soundness:** 1 poor
**Presentation:** 1 poor
**Contribution:** 1 poor
**Rating:** 3
**Confidence:** 2

**Summary:**

The paper presents a method for CNN transfer learning implemented in homomorphic encryption to protect privacy.


**Strengths:**

I'm not aware of the method being implemented in HE before.


**Weaknesses:**

I cannot judge the machine learning aspects, but I don't see a strong novelty on the cryptographic side. The paper claims that some prior work is overly complex without going into details.

I find it concerning that the work relies relatively heavily on non-peer reviewed references by a single author (5 out of 19).

Line 150 says "well-studied by several works" without giving any reference.


Minor issues:
- l6: ::
- l15: .;
- l50: pervacy-persevering (privacy-preserving?)
- l51: diffuclt
- l71: seveal
- l154: After many attempts (unscholarly language)


**Questions:**

Line 108 mentions setting all the weight of an FC layer to zero. Wouldn't that erase all signals?

---

> ### Author Rebuttal · Authors · 2023-08-09
>
> $\textbf{Response}$
>
> We would like to thank the reviewers for their input. Their comments have been thoroughly considered, and altering the manuscript in accordance with these comments will significantly improve the quality of our paper in the next submission.
>
> **C1: The paper claims that some prior work is overly complex without going into details.
> Line 150 says "well-studied by several works" without giving any reference.
>
> A1: Some prior work refers to the works on encoding methods for data (dataset matrices). We should have cited such works we are familiar with.
>
> **C2: I find it concerning that the work relies relatively heavily on non-peer reviewed references by a single author (5 out of 19).
>
> A2: The main techniques in this work did come from the non-peer reviewed references, such as quadratic gradient and volley revolver.
>
> **C3: Line 108 mentions setting all the weight of an FC layer to zero. Wouldn't that erase all signals?
>
> A3: We don't know if setting all the weight of an FC layer to zero would erase all signals or not, but Python simulation experiments suggest, at least for the MLR model,  it works. We did know that we should not set all the weight to zero for neural networks with many layers.
>
> Perhaps, the MLR model is a very simple neural network with only 2 layers. We guess it is the lack of any hidden layer that allows the MLR to work in such a situation.

---

> > ### Comment · Reviewer_ZXqT · 2023-08-16
> >
> > A1: Maybe you could name the missing references now?

---

> > > ### Author Response · Authors · 2023-08-17
> > > **Response to reviewer ZXqT**
> > >
> > > Kim et al. [1] introduced the technique of packing a database matrix into a single ciphertext, and similar approaches have been adopted by other researcher [4]. Jiang et al. [6] discussed the packing method for matrix multiplication. Moreover, numerous studies [3,5] have explored encoding methods for performing CNN inference in the encrypted domain.
> > >
> > > For approximating activation functions using polynomials, Kim et al.[2] employed the least-squares method and provided detailed calculations. In fact, both Python and MATLAB offer a function named "polyfit," which facilitates polynomial approximation in the least-squared sense.
> > >
> > > There are surely other valuable recent contributions in these areas that we have not covered, particularly developments within the past two years. We plan to conduct a comprehensive survey on these topics in our next submission.
> > >
> > > $\textbf{References}$
> > >
> > > [1] Kim, Andrey, et al. "Logistic regression model training based on the approximate homomorphic encryption." BMC medical genomics 11.4 (2018): 23-31.
> > >
> > > [2] Kim, Miran, et al. "Secure logistic regression based on homomorphic encryption: Design and evaluation." JMIR medical informatics 6.2 (2018): e8805.
> > >
> > > [3] Gilad-Bachrach, Ran, et al. "Cryptonets: Applying neural networks to encrypted data with high throughput and accuracy." International conference on machine learning. PMLR, 2016.
> > >
> > > [4] Han, Kyoohyung, et al. "Efficient logistic regression on large encrypted data." Cryptology ePrint Archive (2018).
> > >
> > > [5] Brutzkus, Alon, Ran Gilad-Bachrach, and Oren Elisha. "Low latency privacy preserving inference." International Conference on Machine Learning. PMLR, 2019.
> > >
> > > [6] Jiang, Xiaoqian, et al. "Secure outsourced matrix computation and application to neural networks." Proceedings of the 2018 ACM SIGSAC conference on computer and communications security. 2018.

---

### Official Review · Reviewer_tzCF · 2023-07-06

**Soundness:** 2 fair
**Presentation:** 2 fair
**Contribution:** 2 fair
**Rating:** 3
**Confidence:** 5

**Summary:**

In this paper, the authors proposed a CNN training technique on the homomorphic encryption domain based on transfer learning. A gradient variant called Quadratic Gradient on homomorphic encryption was proposed. And a sigmoid function-based Softmax approximation was proposed. In addition, a new loss function for squared likelihood error was proposed, and a matrix-encoding method called Volly Revolver was also proposed. Finally, they released the code they implemented.

**Strengths:**

Properly implementing functions for training on a homomorphic encryption domain is challenging. It is worth evaluating for implementing this and also disclosing their source code.

**Weaknesses:**

This paper performed a simulation on the MNIST dataset for performance evaluation. This seems too simple a dataset, even considering the homomorphic encryption environment. Although they claim that it is to be the first implementation of transfer learning-based CNN training on the homomorphic encryption domain, a similar study was recently published first. Of course, this paper takes a different approach.

[*] https://openreview.net/forum?id=jJXuL3hQvt

This paper is considered incomplete in several respects. The main reason is that the proposed scheme's threat model needs to be clarified. At first, "the proposed architecture" is not clear. There needs to be a description of the proposed architecture. They should explain the exact part where homomorphic encryption was actually carried out in the transfer learning process and what benefits can be gained from doing so.


**Questions:**

What is the security target to achieve in this paper? It is necessary to explain the threat model that the client or server faces and what benefits the client or server can get from the proposed homomorphic scheme. For example, if the client data is encrypted, the client's privacy can be protected when the server processes the data, but from the server's point of view, it is necessary to clearly explain which portion of transfer learning should be carried out on the encrypted domain. This paper lacks a discussion on this, and it is difficult to know where data processing takes place from the paper alone.

In particular, this paper said that Bootstrapping was not used. It is a very bold and interesting claim. However, they do not provide information on the entire architecture of the proposed scheme. The requirement of bootstrapping is closely related to the information on the homomorphic encryption parameters, such as the ciphertext size, the number of available slots, the number of multiplicative levels, and so on. In fact, the claim that bootstrapping is not used means that homomorphic encryption was applied only in a very small part of the entire operation in their (unknown) architecture, which raises the question of whether the method proposed in this paper is applicable in a practical application. However, judging this part is impossible because it is not explained in detail in the paper.

Finally, [-8,8] is used as the average pooling range. However, it seems too small according to recent work [*].

**Limitations:**

Not exactly.

---

> ### Author Rebuttal · Authors · 2023-08-09
>
> $\textbf{Response}$
>
> We would like to thank the reviewers for their input. Their comments have been thoroughly considered, and altering the manuscript in accordance with these comments will significantly improve the quality of our paper in the next submission.
>
> **C1: What is the security target to achieve in this paper?
>
>
> A1: HE provides a stringent level of security. To use transfer learning,  the client uses a pre-trained CNN model to in advance process their data in order to obtain a new dataset and then encrypt this new dataset. This is the part where transfer learning is used. The server only needs to deal with the calculation of the MLE training over the encrypted data without knowing what exactly it is.
>
> **C2: In particular, this paper said that Bootstrapping was not used. It is a very bold and interesting claim.
>
>
> A2: We would like to use Bootstrapping but didn't due to the current lack of time and funds and the existence of some optimization problems. Just using the Bootstrapping function in HEAAN would consume more time. The homomorphic encryption parameters such as the number of available slots have to be further considered for time optimization. We apologize for not completing the full experiment.
>
> **C3: However, they do not provide information on the entire architecture of the proposed scheme.
>
> A3: The entire architecture of the proposed scheme is actually just the MLR, namely a 2-layer neural network without any hidden layers. For example, in this work, the input layer has 400 nodes plus one constant 1 node and the output layer has 10 nodes.
>
> **C4: Finally, [-8,8] is used as the average pooling range. However, it seems too small according to recent work [*].
>
> A4: The range [-8, 8] is used to generate the polynomial approximate of the Sigmoid function in the output layer. The polynomial approximate developed by the method used in this work has an acceptable performance in a range larger than [-8, 8]. We appreciate the recommendation of the recent work [*].

---

### Official Review · Reviewer_F8vz · 2023-07-10

**Soundness:** 1 poor
**Presentation:** 1 poor
**Contribution:** 1 poor
**Rating:** 2
**Confidence:** 5

**Summary:**

The paper employs a few heuristic methods to accelerate logistic regression training over encrypted data. The heuristics considered include: a new loss function called squared likelihood error (SLE) along with a polynomial approximation of sigmoid function, a faster gradient-descent method based on quadratic gradient, and a matrix encoding method called volley revolver.

**Strengths:**

The only strength of the paper is that it attempts to solve a really challenging problem of learning over encrypted data and it provides the code apriori through an anonymous GitHub link.

**Weaknesses:**

1) First and foremost, the title of the paper is misleading. The paper never deals with CNN training even in the limited context of transfer learning. It is true that most practical ML applications start with a pre-trained model and finetunes the parameters of this model. However, transfer learning implies that the whole model is finetuned apart from learning the application-specific last fully connected (FC) layer. What this paper attempts to do is just learn the last FC layer, which is nothing but multiclass logistic regression (MLR) training. Therefore, the title of the paper should not claim anything about CNN training.

2) Numerous attempts have been made over the last five years attempting to achieve MLR training on encrypted data, which have not been acknowledged in this paper and compared against. For example, see the works starting from:

[A] Crawford et al., "Doing Real Work with FHE: The Case of Logistic Regression", 2018
[B] Han et al., "Logistic regression on homomorphic encrypted data at scale", AAAI 2019
[C] Bergamaschi et al., "Homomorphic Training of 30,000 Logistic Regression Models", 2019

3) This current paper appears to be very similar to the rejected NeurIPS 2022 submission entitled "Privacy-Preserving Logistic Regression Training with A Faster Gradient Variant". While the NeurIPS 2022 submission focused on only the quadratic gradient component, the current paper also introduces the SLE loss. However, it is not clear how this SLE loss function is better. Moreover, what is the expression for the gradient of $ln L_2$ and where is it used in Algorithm 1?

4) The so-called volley revolver does not constitute any novel "matrix-encoding" method. Such packing tricks are regularly used in the context of efficient SIMD operations in FHE.

5) Overall, none of the three claimed contributions (namely, quadratic gradient, SLE loss, and volley revolver) appear to be original or significant enough to make an overall impact.

6) Finally, though the paper claims that the goal is to make logistic regression training practical, not a single experimental result has been shown to prove this point. Running 2 iterations with 128 MNIST images takes approximately 21 minutes and the last line claims that real experiments would take "weeks, if not months". There are other reported works in the literature, which showed more realistic results.

[D] Nandakumar et al., "Towards Deep Neural Network Training on Encrypted Data", CVPR-W 2019
[E] Lou et al., "Glyph: Fast and Accurately Training Deep Neural Networks on Encrypted Data", NeurIPS 2020

**Questions:**

Please see weaknesses of the paper.

**Limitations:**

All the limitations have not been presented and addressed. There appears to be no potential negative societal impact.

---

> ### Author Rebuttal · Authors · 2023-08-09
>
> $\textbf{Response}$
>
> We would like to thank the reviewers for their input. Their comments have been thoroughly considered, and altering the manuscript in accordance with these comments will significantly improve the quality of our paper in the next submission.
>
> **C1: First and foremost, the title of the paper is misleading. Therefore, the title of the paper should not claim anything about CNN training.
>
>
> A1: Yes, we are worried that the title of this manuscript would be seen as deliberately eye-catching, although we think our work is practical for privacy-preserving CNN training based on mere HE. We may change its title or justify it in further submission.
>
> **C2: Numerous attempts have been made over the last five years attempting to achieve MLR training on encrypted data, which have not been acknowledged in this paper and compared against.
>
> A2: Multiple of our own techniques developed before have been used in this work and hence we don't worry to risk plagiarizing others' work. We admit that little survey has been conducted.
>
> **C3: However, it is not clear how this SLE loss function is better. Moreover, what is the expression for the gradient of $\ln L_2$  and where is it used in Algorithm 1?
>
> A3: The new loss function with the Sigmoid function is very HE-friendly and eliminates the need for the conventional Softmax function. We should not have termed this function since it fails to be used in normal neural networks with hidden layers.
>
> The gradient of $\ln L_2$ in this work is a column vector of size [c(1+d)]. In the practical Python programming, we transform the one-column gradient vector into a matrix of size $c \times (1+d)$ and then use the Numpy package to facilitate the computation. In Algorithm 1,   line 23 $(Y-Z)^{\intercal} \times X$ is the matrix that stores the gradient.
>
> **C4: The so-called volley revolver does not constitute any novel "matrix-encoding" method. Such packing tricks are regularly used in the context of efficient SIMD operations in FHE.
>
> A4: The basic idea of volley revolver is to pack the transpose of either matrix for two matrices to perform multiplication. This forms a symmetrical structure between the two packed matrices, which is helpful for both the forward inference and the backward learning. Current solutions for privacy concerns based on HE usually pack matrix into a single ciphertext. We are not sure volley revolver is the first to encode the transpose of a matrix but we did realise it could be used to train neural network in the encrypted domain.
>
> **C5: There are other reported works in the literature, which showed more realistic results.
>
> A5: We appreciate the recommendations. Future changes in this work might introduce the bootstrapping operation to train the MLE model for hundreds of iterations and make some comparisons to these works.

---

> > ### Comment · Reviewer_F8vz · 2023-08-13
> > **Response to Author Rebuttal**
> >
> > Thanks for the rebuttal. After reading through the other reviews and all the responses of the authors, I do not find the responses convincing enough to change my initial rating.

---

> > > ### Author Response · Authors · 2023-08-14
> > > **Response to reviewer F8vz**
> > >
> > > You're welcome for the feedback and thank you for your thoughtful consideration.

---

### Decision · Program_Chairs · 2023-09-21

**Decision:**

Reject

**Comment:**

This paper considers training linear models using FHE. However, it lacks appropriate baselines of comparison as well as a demonstration of the novelty of the proposed methods. The authors are encouraged to include a discussion of these issues, as well as empirical results that more effectively make a case for the practicality of the method.